sustainability

*Article*

# Can Food Safety Practices and Knowledge of Raw Fish Promote Perception of Infection Risk and Safe Consumption Behavior Intentions Related to the Zoonotic Parasite *Anisakis*?

**Uberta Ganucci Cancellieri [1], Giulia Amicone [2], Lavinia Cicero [3], Alessandro Milani [2,*], Oriana Mosca [4], Marialetizia Palomba [5], Simonetta Mattiucci [6] and Marino Bonaiuto [2,7]**

1 Department of Social and Educational Sciences of the Mediterranean Area, University for Foreigners "Dante Alighieri" of Reggio Calabria, 89125 Reggio Calabria, Italy; ganucci@unistrada.it
2 Department of Psychology of Development and Socialization Processes, Sapienza University of Rome, 00185 Rome, Italy
3 Faculty of Psychology, e-Campus Telematic University, 22060 Novedrate, Italy
4 Department of Education, Psychology, Philosophy, University of Cagliari, 09123 Cagliari, Italy
5 Department of Ecological and Biological Sciences, Tuscia University, 01100 Viterbo, Italy
6 Department of Public Health and Infectious Diseases, University Hospital "Policlinico Umberto I", Sapienza University of Rome, 00185 Rome, Italy
7 CIRPA–Interuniversity Research Centre in Environmental Psychology, Sapienza University of Rome, 00185 Rome, Italy
* Correspondence: a.milani@uniroma1.it

check for updates

**Citation:** Ganucci Cancellieri, U.; Amicone, G.; Cicero, L.; Milani, A.; Mosca, O.; Palomba, M.; Mattiucci, S.; Bonaiuto, M. Can Food Safety Practices and Knowledge of Raw Fish Promote Perception of Infection Risk and Safe Consumption Behavior Intentions Related to the Zoonotic Parasite *Anisakis*? *Sustainability* **2023**, *15*, 7383. https://doi.org/10.3390/su15097383

Academic Editor: Marian Rizov

Received: 23 March 2023
Revised: 24 April 2023
Accepted: 25 April 2023
Published: 28 April 2023



**Abstract:** The study of the zoonotic parasites of the genus *Anisakis* and human anisakiasis is an increasingly hot topic in evolutionary biology and epidemiological studies carried out on natural and accidental (human) hosts, given the risk of this parasite to human health. However, the assessment of social-psychological factors relevant to *Anisakis*' risky consumption of human behavior is still an understudied topic. Given the centrality of the topic, highlighted by its presence in Goals 2 (subgoal 2.1, achieve food security and improve safe nutrition) and 3 (health and well-being) of the 2030 Agenda, it appears necessary to deepen our social-psychological knowledge regarding this specific topic. There is plenty of psychological research focused on antecedents of fish and seafood consumption; however, parasite risk is not often specifically examined. This research is aimed at increasing the safety of consumers' seafood products by examining their psychological aspects, such as knowledge, perception, awareness, and concern about *Anisakis*. Past and future behavior intentions were also investigated. Analyses were carried out on a sample of 251 subjects, and a path analysis model was used to explain possible relations assumed among the variables. The results of the study showed that habits related to raw fish consumption and self-perceived health were, respectively, positively, and negatively correlated with a higher perceived risk of contracting anisakiasis. This perceived risk in turn correlates positively with a greater willingness to pay for *Anisakis*-free fish. Similarly, prior knowledge of the disease was found to be associated with prior avoidance of fish consumption, which in turn was found to be positively correlated with a greater willingness to pay for *Anisakis*-free fish.

**Keywords:** *Anisakis*; risk perception; knowledge; health; habits; food security; willingness to pay

## 1. Introduction

The present study draws its aim from the results achieved in Italy in the frame of a European research project entitled "PARASITE" (Risk assessment with integrated tools in EU fish production value chains) which concerned the distribution and epidemiology of anisakid nematodes, parasites in the wild fishery from European waters. Regarding fishery products, the most hazardous parasite species recognized by the EFSA are the anisakid nematodes, particularly those of the genus *Anisakis* [1].

Different species of *Anisakis* have a complex life cycle involving various marine organisms at different levels of the trophic web of the marine ecosystem (Figure 1). Crustaceans (krill) are the first intermediate hosts, while fish and squid are the intermediate/paratenic hosts and, finally, marine mammals, mainly cetaceans, are the definitive ones where the nematodes reach the adult stage (Figure 1). The third-stage larvae of *Anisakis* can infect the edible part of numerous fish and squid species worldwide, including those of commercial importance (i.e., anchovies, hakes) (for a review, see [2]). Humans become accidental hosts by eating raw or undercooked parasitized fish or squid containing live larvae of these parasite species. *Anisakis* penetrates the mucous layers of the gastric-intestinal tract, causing the zoonotic disease known as "anisakiasis" ([2] for a review). Therefore, the zoonotic disease is acquired by humans through the ingestion of raw or undercooked sea products [3]. Thus, the study of *Anisakis* is nowadays an increasingly hot topic in medicine, biology, and epidemiological studies. Among the genetically characterized species of the genus *Anisakis*, so far *A. pegreffii* and *A. simplex* (s.s.) are known to be able to provoke human anisakiasis [2–4]. Those two zoonotic species can provoke in humans gastric, gastro-allergic, and intestinal anisakiasis. In addition, IgE hypersensitization in Italian patients has been also reported due to the species *A. pegreffii* [2]. Anisakiasis is becoming an increasingly significant human health risk which must be made known to the population, due to the growing increase in this severe infection, particularly in countries where raw fish consumption is frequent (reviewed in [2,3,5]). As a result, cases of the disease are increasingly being reported from Asia (more than 50%), with the remainder coming from Europe (i.e., Italy, Spain, France, and Croatia) [2,6,7]. Therefore, in these countries, where fish and raw fish consumption are a central part of the diet, this phenomenon cannot be underestimated.

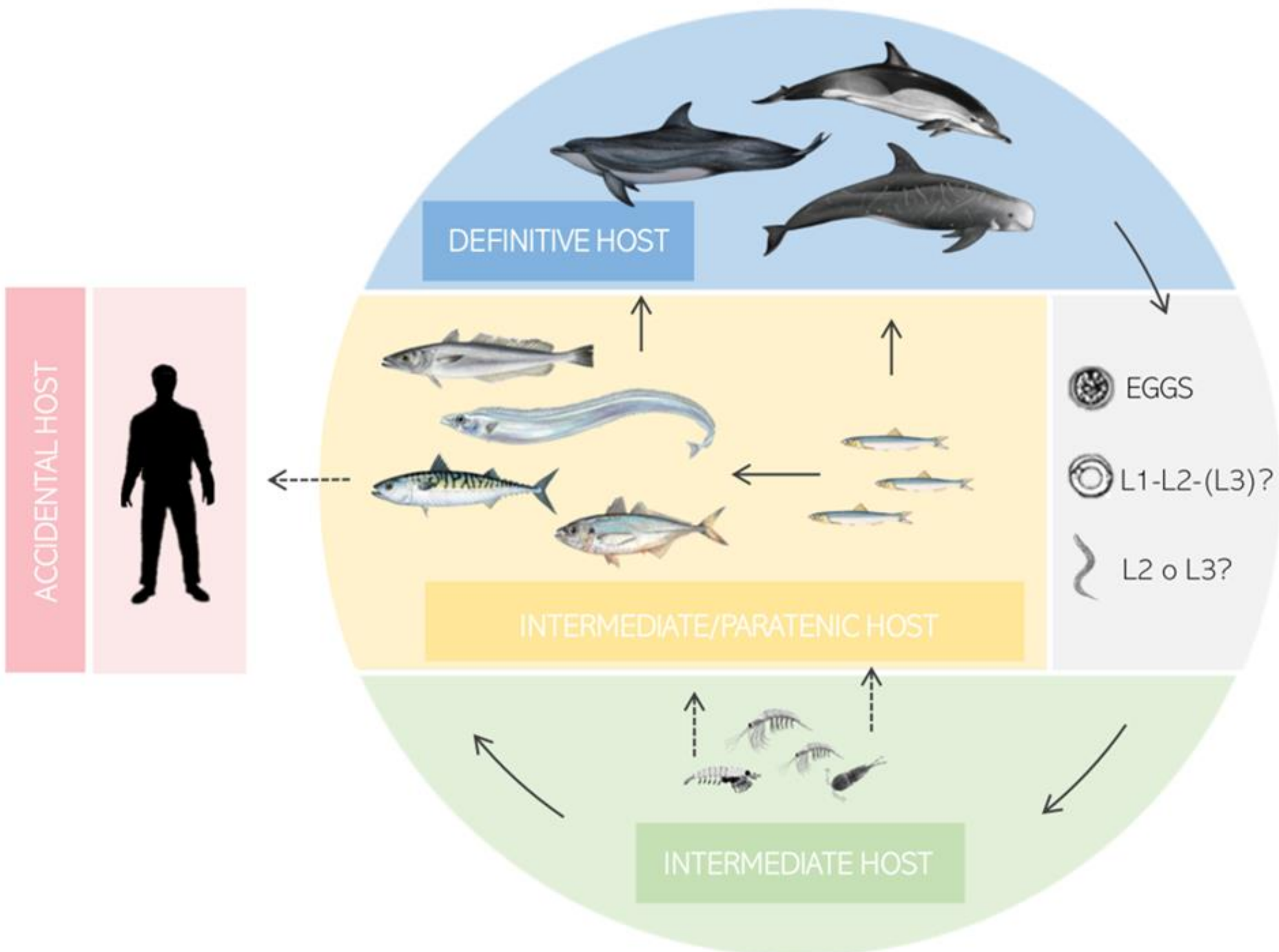


**Figure 1.** Schematic general life cycle of the zoonotic parasite species, *Anisakis pegreffii* and *A. simplex* (s.s.). Note. L1-L2-L3 refers to Anisakis larval stages.

Given the severity of those symptoms, the generally low awareness of this parasite, and its global interest [1], it becomes useful to investigate the population's risk perception and knowledge of *Anisakis* (and the respective zoonotic disease anisakiasis) [6].

The centrality of the issue at both the individual and community levels, in recent years, has also been addressed by the "Sustainable Development Goals" in the United Nations

2030 Agenda [8]. There are two goals in the 2030 Agenda that specifically consider this phenomenon.

First, Goal 2 (subgoal 2.1, "Achieve food security and improve healthy nutrition") declares "By 2030, eliminate hunger and ensure access to safe, nutritious and sufficient food for all people, particularly the poor and people in vulnerable situations, including children, throughout the year" [8]. To achieve this goal, it is crucial to try to provide the appropriate information to the global population, especially the poorest and least educated, about possible disorders or infections due to risky eating behaviors. In particular, concerning the risk of contracting anisakiasis, it is especially the Asian and oceanic populations that are at greatest risk, in areas where hunger and poverty often rage [2,6,7].

In line with Goal 2, Goal 3 of the 2030 Agenda (health and well-being) also seems to assume centrality in the issue. Particularly, disease (such as anisakiasis) prevention is one of the key subgoals (3d).

Therefore, given the high risk associated with this parasite and the anisakiasis disease [2–4], and given the low perception of risk and lack of knowledge about this pathology, it is of paramount importance—in parallel with the development of medical and biology research—to identify the social-psychological characteristics that help prevent specific diseases, promoting campaigns to raise the awareness of this nematode parasite, as well as the risks associated with the consumption of unabated raw fish.

Indeed, the assessment of social-psychological factors relevant to *Anisakis*' risky consumption behaviors is still an understudied topic, except for a few rare—basically Iberians—studies [9,10]. Plenty of psychological research has been focused on antecedents of fish and seafood consumption: the perception of physical and health risks is one of the barriers to that consumption. However, the physical and health risks considered in these studies are mainly related to the risk of eating contaminated fish and seafood (mainly by pollution agents); parasite risk is usually mentioned but not specifically examined ([11,12] for a review).

Parasite-related risk perception and behaviors have indeed been studied about the consumption of foodstuffs of animal origins. However, these studies have mainly targeted domestic food preparation [13] with special attention to meat, eggs, and milk. On the contrary, fish has only been marginally targeted by such studies. The only survey-based Contingent Valuation (CV) studies to investigate consumer Willingness to Pay (WTP) for *Anisakis*-free fish and to analyze consumer responses to the presence of *Anisakis* in fishery products in the literature are the aforementioned Iberian studies [9,10]. However, these studies have focused on descriptive analyses (frequencies of questionnaire responses in individual questions) and only marginally investigated the correlation of several independent variables on the dependent variable (WTP) [9].

Due to the epidemics that have occurred over the past decades (e.g., avian, swine, mad cow disease) dietary behaviors related to such foods have been investigated [14–16], while there is a tendency to underestimate the risk associated with the consumption of raw fish. Moreover, given the increasing trend, with the influence of Asian cuisine, of eating sushi and other types of raw fish [17,18], the risk of contracting anisakiasis has increased sporadically. It, therefore, becomes not only necessary to inform people about these risks but also to understand how to promote healthy eating habits, a goal that psychology should ideally achieve.

Hence, the purpose of the present research is to represent a first step toward a broader understanding of the social-psychological determinants that lead to the uncontrolled consumption of raw fish and subsequently to understand the determinants of behaviors that can reduce these risks. To achieve this goal, this research seeks to fill a gap in the literature about risk perception and knowledge of *Anisakis*. With this study, sequential correlations between different social-psychological variables that can predict correct behaviors to reduce the spread of this disease were investigated for the first time.

As mentioned above, only one study investigated the correlation between some independent variables and WTP, but it was limited to the analysis of a small number of

independent variables and did not assume any kind of sequentiality among the variables examined [9]. Finally, this is the first study to investigate the relationships between these variables in an Italian context and sample (at high risk of contracting anisakiasis).

### *1.1. How Perceived Health, Eating Habits, and Food Risk Perception Influence Our Eating Choices*

Several studies in the literature highlight how quality of life and health perception can significantly influence people's moods, emotions, and even behaviors [19–21]. This is because perceiving ourselves as healthy allows us not to focus on what may be perceived as risky for our health [22].

Classical research on risk perception about self-perceived health [23,24] has also shown how these two constructs influence each other. In fact, these studies have shown how particular situations or self-perceptions can influence our risk perception concerning certain phenomena. In particular, it is clear in the literature that the perception of a situation as risky depends on social and individual factors [23]. As for individual factors, the perception of one's health, as well as other factors related to it (e.g., age or gender) [25] seem to have a significant impact on individual risk perception. Finally, Wildavsky and Dake [26], in their paper on theories of risk perception, also identified several individual factors that can influence perceived risk. In particular, with regard to risks related to our health, individual perceptions of one's health seem to significantly change the perception of a situation as risky or not. As with any other context/behavior, a poor health perception increases a sense of fragility, which leads to feeling more vulnerable and perceiving any action as riskier [22].

Similarly, concerning our nutrition, not perceiving ourselves as being in excellent health will lead us to adopt healthier behaviors, such as avoiding alcohol or foods that are harmful or potentially harmful to our health, as they are seen as riskier [27,28]. These foods can include both those that have a detrimental impact on our bodies in terms of nutritional values and those that could be contaminated (by chemicals in the environment or by possible parasites). Among the foods that may have contaminants, fish can be considered to be subjected to chemical contaminants (e.g., mercury) and parasites (e.g., *Anisakis*). Indeed, two studies conducted in Australia and the United States [11,29] have shown that sea product consumers are concerned about the presence of mercury and, to a lesser extent, of other contaminants such as polychlorinated biphenyls in fish. Other studies strongly emphasize the problem of *Anisakis*, which impacts not only the health of the population but also the economy [30].

Such contaminants make fish, as well as most foods of animal origin, not only a fundamental source of sustenance and nutrition but also potentially risky to health, with non-negligible risks from the population [11,29,31–33]. This risk rate tends to increase if such foods are consumed raw or undercooked. In fact, as in the case of other foods (e.g., meat), cooking them reduces the risk of assimilating most of the organic contaminants present in food [1,34].

Since fish is a key component of human nutrition, and considering that "*all wild caught seawater and freshwater fish must be considered at risk of containing any viable parasites of human health concern if these products are to be eaten raw or almost raw*" [1], it is essential to try to develop effective prevention methods without having to eliminate the consumption of sea products from our diet. Currently, the only effective processing methods to kill *Anisakis* are freezing at −20 °C for 24–48 h, or cooking fish products at more than 60 °C [1]. A more recent study [35] evaluated traditional open-air drying as an alternative and effective treatment for inactivating *Anisakis*. These preparations and basic fish consumption habits should prevent large-scale new allergic episodes even in already-sensitized individuals [36]. However, the methods just described inevitably alter the texture and flavor of the fish. No known method of anisakid removal does not result in a substantial or momentary (freezing) chemical alteration of the fish.

Several studies have already related health perception to fish consumption [12] and the results show that health perception is negatively related to sea product consumption.

The study by Bao et al. [9] shows how, even concerning fish and raw fish consumption, the participants who perceived themselves to have worse health, perceived high fish-related risk and were also the least likely to consume fish and raw fish. This phenomenon, as highlighted above, is driven by an increase in risk perception by people with poor health. Perceiving raw fish as risky may therefore lead to the avoidance of raw fish consumption in people with a poor perception of their health. Perceiving fish consumption as highly risky will then lead these people to be willing to pay a higher price to override that risk. As a consequence, the people who perceive a higher risk of contracting anisakiasis are the same people who are willing to pay more for *Anisakis*-free fish [9]. Confirming this, numerous studies have shown risk perception to be a crucial predictor of behavioral intention [37,38], especially with regard to behaviors related to one's health. In particular, risk perception has been identified as the motivation to act and is thus closely related to our intentions [39].

Regarding habits, these have been defined by Ajzen [40] as behaviors that are regularly repeated and tend to occur countless times, initially driven by individuals' motivation until they recur on an automatic level, which is why it is crucial to study them to predict and influence people's future behaviors. With this purpose, several theoretical models have been related to habits and behavioral intentions over the years, such as the theory of interpersonal behavior [41]. Furthermore, as habits are considered behaviors that are repeated over time [40], it has been seen that these are one of the main predictors of future behavior [42].

Examining eating behavior and fish consumption, Honkanen et al. [43] found that habit is a strong predictor of the intention to consume fish, and similarly, Juhl and Poulsen [44] and Verbeke and Vackier [45] identified fish consumption habits as a strong predictor of fish consumption frequency. Confirming the results of the research just cited, several studies [46–50] have shown how high fish consumption is mostly the expression of a cognitive representation of a pre-existing habit performed with a lack of awareness and control, rather than the result of reasoning. Considering then that old habits are difficult to break and new ones to form, consumers' choice of fish is strongly influenced by habits that emerge and are reinforced based on past satisfactory/unsatisfactory experiences associated with the same behavior [51].

It is so conceivable that increased experience regarding fish consumption (due to habits) leads consumers to eat more fish and consequently to perceive a higher risk associated with eating fish and raw fish. In fact, past experience with a risky event, as demonstrated by Thistlethwaite et al. [52] regarding flooding, increases the perception of risk and the implementation of appropriate behaviors to decrease it. Similarly, regarding the relationship between habits and risk perception, the well-known study by Sitkin and Pablo [53] showed that past behavior and familiarity with certain issues increase their perception of riskiness.

### *1.2. How Anisakis' Experience Drives Consumption of Fish and Raw Fish: The Impact of Previous Knowledge and Behaviors on Future Behavioral Intentions*

There are several studies in the literature that have related knowledge of a particular object/phenomenon to the subsequent intention to act toward that object/phenomenon. In particular, such pieces of evidence have demonstrated the impact of knowledge on our subsequent behavioral intentions [54]. Other classical studies [37] have also shown how cognition and knowledge of a given topic and its issues are the prerequisites to developing a strong intention to act.

Therefore, considering knowledge of a given phenomenon as one of the prerequisites for people to behave properly to cope with that [55], a lack of understanding and knowledge can only be perceived as an obstacle to the correct response to the phenomenon [56]. It has been shown that appropriate knowledge not only encourages people to act but also facilitates the implementation of the appropriate behaviors to cope with or avoid that phenomenon [57]. An example of such behaviors might be, for example, an increased WTP for *Anisakis*-free fish.

Over the past few years, because of the COVID-19 pandemic, there has been a realization of the importance of accurate knowledge of prevention methods for this disease. A large body of literature that has emphasized the crucial role of knowledge in the prevention and control of infectious diseases (especially COVID-19) has also developed [58]. Based on these studies, it is therefore conceivable that adequate knowledge is also crucial for the prevention of other infectious diseases, such as anisakiasis.

The study by Baptista et al. [59] analyzed how knowledge of the risks associated with seafood consumption can influence the eating practices of its consumers, leading people to avoid risky behaviors. That study found that there is not a high level of knowledge in the population about the potential risks associated with seafood consumption and that such knowledge succeeds in guiding people toward implementing appropriate behaviors and becoming more willing to engage in them. In fact, the authors emphasized that it is necessary to be able to create a chain of knowledge from the producer to the consumer to prevent risky practices. The study by Bao et al. [9] also showed how the lack of knowledge regarding anisakiasis prevention methods suggests a significant potential risk of anisakiasis and allergic episodes for raw fish consumers.

Therefore, efforts should be made to promote knowledge of raw fish consumers' proper dietary practices, who may be at risk of contracting anisakiasis by not knowing this disease and how to avoid it.

If healthy practices of fish and raw fish consumption are to be promoted, however, it cannot be assumed that focusing on knowledge is the only way forward. Indeed, promoting knowledge about a particular phenomenon does not inevitably lead people to enact the right behaviors, but only increases the likelihood that they will do so [57].

As for the relationship between past behavior and behavioral intention, several studies have indicated that past behavior and related experience are the focal predictors of future behavioral intention [42,60,61]. Depending then on the past behavior enacted and the positive or negative responses related to the same behavior, this will influence subsequent behavioral intention differently [60,61].

### 1.3. Hypotheses

Based on the above studies and theoretical reasoning, six hypotheses drive the present research.

The hypotheses are formulated as follows and presented in Figure 2:

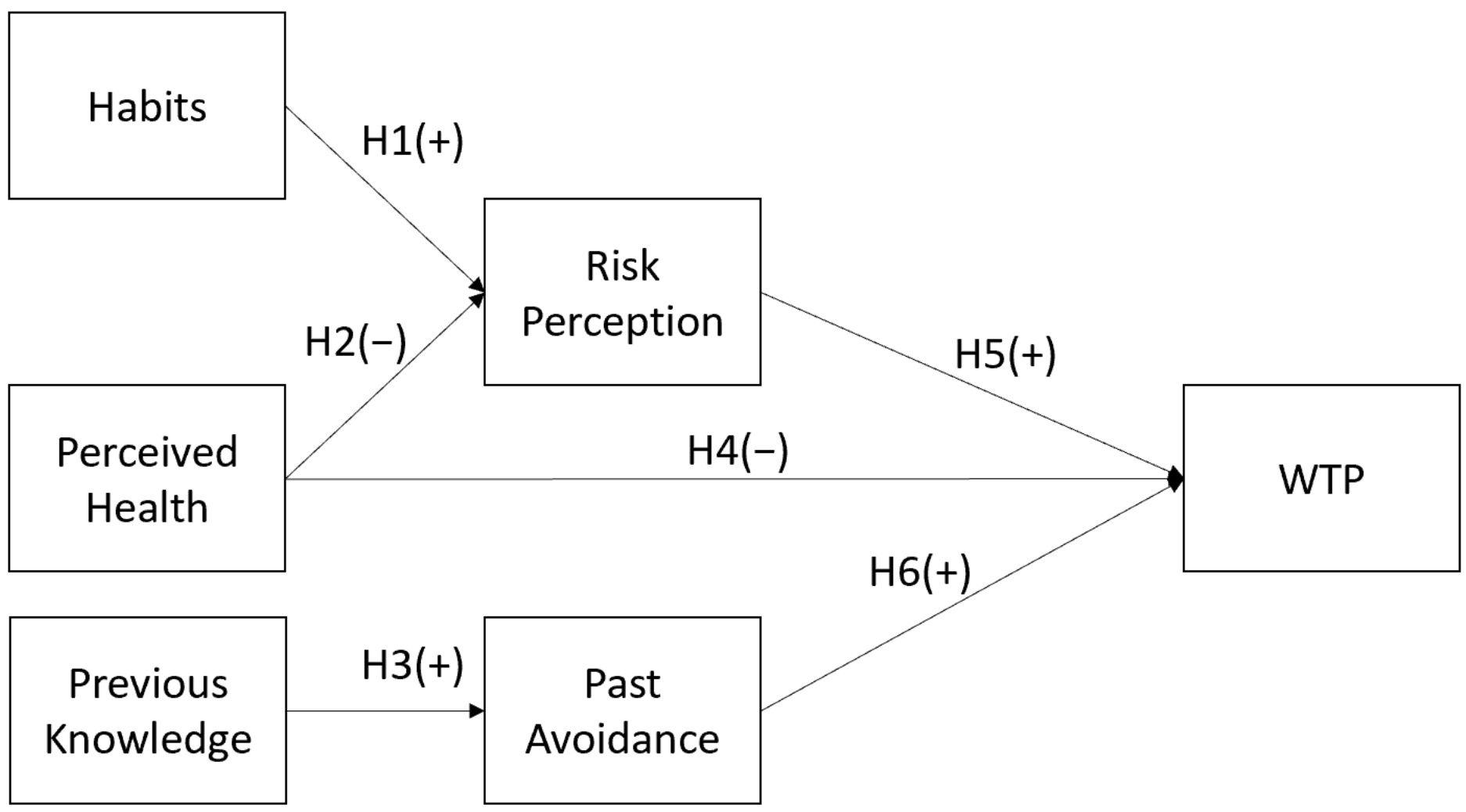


**Figure 2.** Hypothesized path model.

**H1:** *"Habits", namely raw fish consumption habits, are positively related to "Risk perception", namely risk perception to contract anisakiasis.*

**H2:** *"Perceived Health" is negatively related to "Risk perception", namely risk perception to contract anisakiasis.*

**H3:** *"Previous Knowledge", namely the perceived prior knowledge of Anisakis, is positively related to "Past Avoidance", namely past avoidance of raw fish intake because of Anisakis.*

**H4:** *"Perceived Health" is negatively related to "Willingness to pay", namely willingness to pay for Anisakis-free fish.*

**H5:** *"Risk Perception", namely risk perception to contract anisakiasis, is positively related to "Willingness to pay", namely willingness to pay for Anisakis-free fish.*

**H6:** *"Past Avoidance", namely past avoidance of raw fish intake because of Anisakis, is positively related to "Willingness to pay", namely willingness to pay for Anisakis-free fish.*

These hypotheses were tested and presented throughout the article.

Specifically, the following sections present the materials and methods used in conducting the study, the results that emerged from the tested path model, the discussions, and the conclusions developed from the findings. At the end of the article, the limitations and possible prospects of the present research are then presented.

## 2. Materials and Methods

### *2.1. Procedure, Sample, and Materials*

This research aimed to investigate the perception of foods concerning the risk of consuming them. Specifically, in this study, people's eating habits regarding raw fish consumption and the perception and risk assessment of parasites in foods (*Anisakis* in raw fish) were investigated. The study was conducted in 2017 (August–September) through an online survey in a sample of Italian citizens, who voluntarily filled out the questionnaire. From the original sample (N = 266), those who did not respond to questions related to *Anisakis'* knowledge and scales on consumption habits were removed. In addition to conducting statistical checks and analyses of the data, the quality of the data was monitored reporting—if necessary, eliminating—participants who took an unusually short or long time to complete the study or who reported unlikely response patterns [62]. The final sample (N = 251) is composed of 74 (29.5%) men and 177 (70.5%) women and ranges in age from 18 to 78 years, with an average age of 33 years. Participants were recruited via online sharing (using various social networks) of the link (e.g., via mail) to the questionnaire. Specifically, a convenience sample of the population composed of participants who had not developed an *Anisakis* infection was chosen to be investigated ([63] for a review).

The questionnaire consists of two parts, the first part in which questions are presented in a canonical manner and a second part in which questions are presented following a different survey-based methodology. Such a differentiation initially consists of providing various forms of information regarding the topic of *Anisakis* (how to contract it, the severity of anisakiasis disease, the likelihood of contracting the disease, and methods of prevention) and then presenting a scenario in which a 100% effective *Anisakis* removal method is imagined (Contingent Valuation; CV). CV is a survey-based methodology for assigning monetary values to nonmarket resources [64,65], which has its roots in random utility theory [66,67], introducing certain scenarios to then require the willingness to pay for a certain product/service.

The measurement variables examined in our study are as follows.

"Perceived Health", a single purpose-built item: "How would you rate your health?"; its response scale is a 4-point Likert scale (from 1 "weak" to 4 "excellent").

"Risk Perception", namely risk perception to contract anisakiasis, a single purpose-built item: "Consider the *Anisakis* a health risk. What do you think are the chances that you

will develop Anisakiasis and/or an *Anisakis* allergy in the next 10 years?"; the response scale is a 5-point Likert scale (from 0 "no chance" to 4 "absolute certainty").

"Willingness to Pay", namely willingness to pay for *Anisakis*-free fish, a single item constructed according to the CV method introduced by the following scenario:

"Imagine that a treatment has been discovered that guarantees the removal of the *Anisakis* worm (and related allergens) from fish. Assume that this treatment is 100% effective and that there is no effect on the quality and taste of the fish. Considering that 1 kg of anchovies costs an average of 4.50 EUR per kg, indicate the amount you would be willing to pay for 1 kg of anchovies subjected to the treatment that removes *Anisakis* and its allergens".

The response possibilities to this scenario are 10 price units (from 0 EUR to 9.00 EUR). These price units were converted into scores from 1 to 10, this variable being an equivalent interval variable (between each scale's score there is a price increment equal to 0.90 EUR).

"Previous Knowledge", namely prior subjective knowledge of *Anisakis*, a purpose-built variable from 2 items ("Have you ever heard of *Anisakis* or herring worm or cod worm or parasites in fish? and "Do you know the prevention methods against *Anisakis* or worms in fish?"). These two items are dichotomous variables with yes/no response options. "Yes" responses were given a score of 1 while "No" responses were given a score of 0. The scores of the variable "Previous Knowledge" are given by the sum of the scores of its two component items. This variable thus ranges from 0 (answer "No" to both questions) to 2 (answer "Yes" to both questions).

"Past Avoidance", namely past avoidance of raw fish intake because of *Anisakis*, a purpose-built variable calculated following a follow-up question that reads "Have you ever avoided buying or eating any kind of fish because of the presence of *Anisakis*?" with a yes/no answer. The question reads "If you answered Yes to the previous question, what kind of fish did you avoid? Multiple possible options". The answer set of this variable consists of 12 species of fish. The value of this variable is given by the sum of the species avoided, with potential values ranging from 0 (never avoided eating or buying fish because of the risk of *Anisakis*) to 12 (if you avoided all the species mentioned).

"Habits", namely raw fish consumption habits, a single purpose-built item that reads "If you eat raw fish at friends' or acquaintances' houses or in restaurants and public places what kind of dishes do you eat?" The score of this variable ranges from 0 to 6 and depends on the types of raw fish eaten (from none to all those in the list). An example of a response is "Raw fish with lemon and vinegar".

The variables, whose score is given by the sum of the different items' scores, consider such items as formative indicators [68,69].

### 2.2. Data Analyses

Preliminary analyses (data exploration and reliability) were oriented to explore the sample and measurement scales: these were carried out with the open-source software "Jamovi 2.3.24". The reliability analysis was tested on the only variable on which it could be calculated ("Previous Knowledge") through the item's tetrachoric correlation. The analyses showed a large correlation (rphi = 0.89) following Agresti's guidelines [70].

The correlation matrix between the variables examined is shown below (Table 1).

**Table 1.** Correlation Matrix.

| | PH | CH | PK | PA | RP | WTP |
|---|---|---|---|---|---|---|
| Perceived Health (PH) | — | | | | | |
| Consumption Habit (CH) | 0.09 | — | | | | |
| Previous Knowledge (PK) | 0.14 * | 0.16 * | — | | | |
| Past Avoidance (PA) | 0.07 | 0.06 | 0.31 *** | — | | |
| Risk Perception (RP) | −0.13 * | 0.19 ** | −0.01 | −0.04 | — | |
| WTP | −0.12 | 0.07 | 0.08 | 0.20 ** | 0.15 * | — |

Note. * $p < 0.05$, ** $p < 0.01$, *** $p < 0.001$.

The correlation matrix shows that there are no excessively high correlations, and thus no multicollinearity ($r < 0.70$) among the variables investigated in the model.

The same software was applied for all hypotheses by testing a path model through the "PATHj Package" [71] for path analysis, using the Maximum Likelihood (ML) method to estimate model parameters with the calculation of standard errors based on the observed information matrix. In addition to the significance of the $\chi^2$ value considered for assessing the overall fit of the model ([72], for a detailed account of its weak reliability), it was also considered the more reliable ratio between $\chi^2$ and degrees of freedom (under 3 being the threshold acceptability) according to McIver and Carmines [73]. Other indices were considered to have a more comprehensive evaluation [74], such as the comparative fit index (CFI) and the Tucker–Lewis index (TLI), that compare the factor model's fit of a baseline model in which all observed variables are expected to be uncorrelated. CFI and TLI > 0.90 indicate an acceptable fit, while values greater than 0.95 indicate a good fit. The standardized root mean square residual (SRMR) is a residual-based statistic for which values less than 0.10 and 0.05 are considered, respectively, acceptable and as a good fit. The root mean square error of approximation (RMSEA) measures the difference between the model-implied covariance matrix and the population matrix to control sampling variability. RMSEA values of 0.05 or less indicate a close fit, and values up to 0.08 represent a reasonable error of approximation.

## 3. Results

In testing the hypotheses, as shown in Table 2, the model yielded optimal fit indices, including the $\chi^2$ and $\chi^2/df$ ratio.

**Table 2.** Model Fit Indices.

| χ2 | df | *p* | χ2/df Ratio | CFI | TLI | RMSEA | SRMR |
|---|---|---|---|---|---|---|---|
| 7.53 | 7 | 0.38 | 1.08 | 0.99 | 0.98 | 0.02 | 0.04 |

Looking at the structural coefficients (Figure 3):

In testing H1, it emerges that "Risk Perception" is predicted positively by "Habits" ($\beta = 0.20$, $p < 0.001$);

In testing H2, it emerges that "Risk Perception" is predicted negatively by "Perceived Health" ($\beta = -0.15$, $p < 0.05$);

In testing H3, it emerges that "Past Avoidance" is predicted positively by "Previous Knowledge" ($\beta = 0.31$, $p < 0.001$);

In testing H4, it emerges that "Willingness to pay" is predicted negatively by "Perceived Health" ($\beta = -0.12$, $p < 0.05$);

In testing H5, it emerges that "Willingness to pay" is predicted positively by "Risk Perception" ($\beta = 0.14$, $p < 0.05$);

In testing H6, it emerges that "Willingness to pay" is predicted positively by "Past Avoidance" ($\beta = 0.22$, $p < 0.001$).

As shown in Figure 3, all hypotheses are confirmed, and therefore all six alternative hypotheses are accepted. Finally, the path model explains the following proportion of variance of the three endogenous variables, i.e., "Risk Perception", "Past Avoidance", and "Willingness to pay" (6%, 10%, and 8% of the accounted variance, respectively).

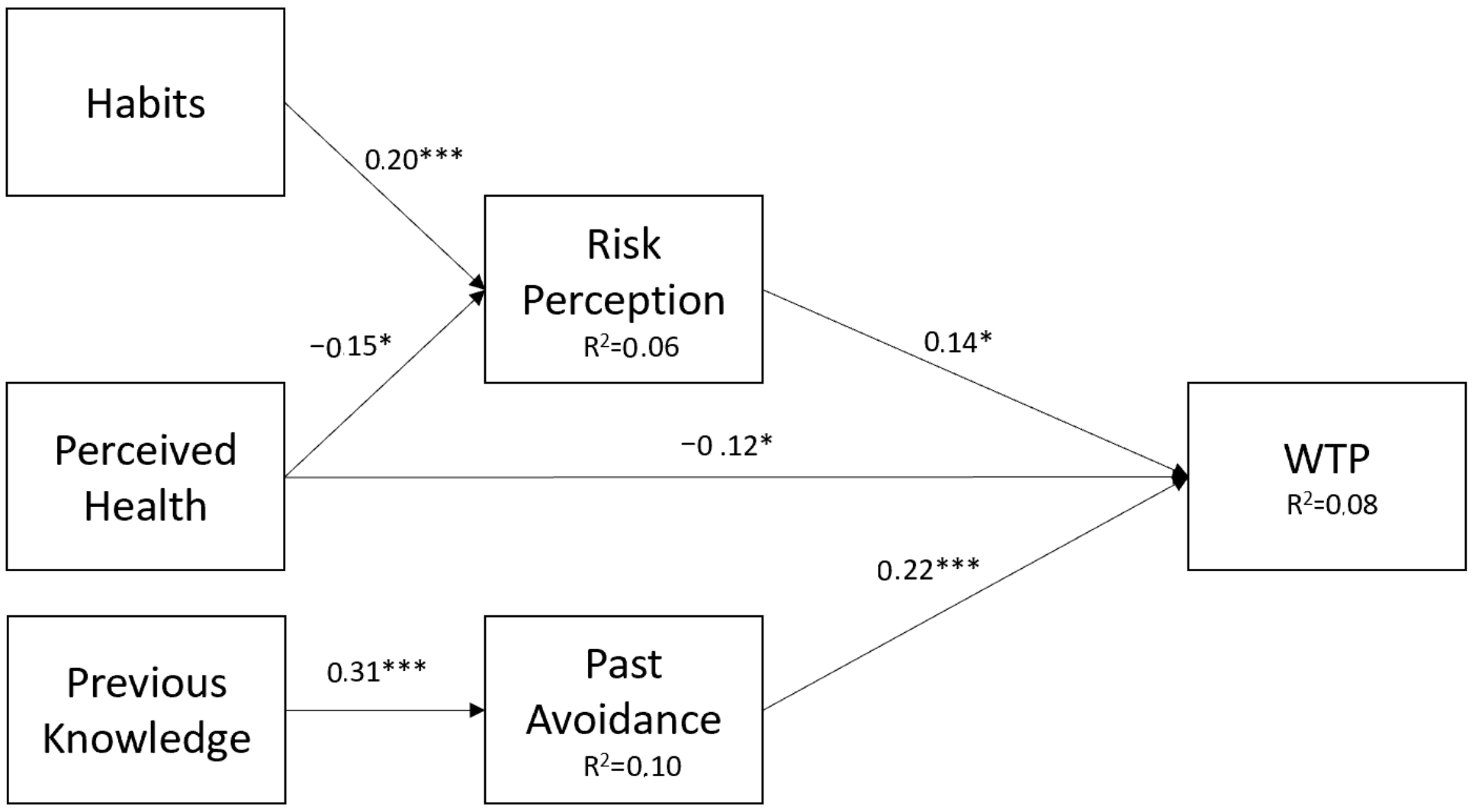


**Figure 3.** Path model and hypotheses testing. Note. * $p < 0.05$, *** $p < 0.001$.

## 4. Discussion

The results show that perceived health negatively and significantly predicts the perceived risk of contracting anisakiasis. This is consistent with the research of Koo and Boo [75], which reveals that poor health perception is positively correlated with a higher perceived risk of cardiovascular diseases. Similarly, the older study by Fiandt et al. [76] also highlights that poor health perception is correlated with a higher risk perception of chronic diseases.

A higher risk perception of contracting anisakiasis will therefore also be predicted by a poor health perception.

Thus, it can be inferred that people who perceive that they are not in excellent health will have a riskier perception of things around them, including the food they eat daily. The results also reveal how individuals who have a higher risk perception of contracting anisakiasis are the same individuals who will be more willing to pay an extra price for *Anisakis*-free fish. There are several studies in the literature that have analyzed the impact of the perception of a given risk on the willingness to pay to reduce or cancel it [77–80]. They all lead to the same conclusion: perceiving something as risky leads people to be willing to pay to reduce or cancel that risk, especially if it directly impacts their health.

Similarly, it is easy to understand how a worse perception of one's health leads respondents to be more likely to pay an additional price for *Anisakis*-free fish. A poor perception of one's health, as just described, by increasing the perception of the risk of contracting anisakiasis, will lead people to act, even at the cost of paying a higher amount, to minimize that risk.

The only study that has previously investigated the social-psychological determinants of WTP for *Anisakis*-free fish [9] did not analyze possible relationships between perceived health and perceived risk of contracting anisakiasis, while a possible relationship between perceived health and WTP was analyzed, but this was not found to be significant. Instead, the results confirm the study by Bao et al. [9] in reference to the positive relationship between the perceived risk of contracting anisakiasis and WTP.

It then emerged how the prior subjective knowledge of *Anisakis* positively predicts past fish avoidance behavior due to *Anisakis* and how this behavior subsequently predicts willingness to pay for *Anisakis*-free fish.

This means that people who are most familiar with such parasite will be the most likely to have avoided eating fish in the past, because of *Anisakis*. These will then be the same people who will pay the most for *Anisakis*-free fish. In fact, people who do not have adequate prior knowledge of *Anisakis* and how to prevent anisakiasis will have no reason to avoid eating fish or raw fish, not knowing the potential risks, as also demonstrated by

Nardi et al. [81] in the study of food risk perception. Finally, people who have avoided eating fish in the past, because of the risk of the parasite, will be the ones willing to pay more for healthy fish. This phenomenon can be interpreted as a willingness of these people to not avoid consuming fish again because of the risk of this parasite.

The study by Bao et al. [9] did not examine the relationship between prior knowledge and past avoidance. On the other hand, as for the relationship between prior knowledge and WTP, this was analyzed and found not to be significant, while the results of the present study confirm the previous research of Bao et al. [9] by highlighting how past behavior is a focal predictor of future intention to pay for *Anisakis*-free fish.

Finally, the results showed how people with greater experience of fish consumption and who eat a greater number of fish species will perceive a greater risk of contracting anisakiasis. This could be since people who have eaten and eat more fish feel more at risk than those who have eaten and currently eat less fish. This perception of risk is probably not due to a perception of fish as riskier by these individuals, which would likely lead them to avoid fish. These persons, it can be assumed, perceive a greater risk of contracting anisakiasis because of the increased likelihood (in percentage terms) of eating an infected fish, as also shown by Dickson-Spillmann et al. [82] concerning the consumption of contaminated food. These results are indeed consistent with the study by Kim et al. [83], which exposes how past experience influences the risk perception of raw or undercooked foods.

Again, the study by Bao et al. [9] did not analyze the relationship between habits and risk perception. On the other hand, as previously mentioned, the relationship between the perceived risk of contracting anisakiasis and WTP for *Anisakis*-free fish was also found to be significant and positive in this previous study.

The results are particularly innovative, as no sequential regression models, that took into account the variables examined, had previously been developed in food-related research and more specifically for the *Anisakis* disease. The research is also innovative because of the subject matter. In fact, in the literature, no other studies were found that addressed the issues of *Anisakis* from such a specific social-psychological perspective. Even with reference to the only study that had previously investigated some social-psychological determinants of intention to pay for *Anisakis*-free fish [9], a huge step forward has been made. This is made more evident by the results that emerged from the present study, which not only filled in some previous gaps due to not considering certain relationships between variables but also revealed important effects that had not emerged as significant in the previous study. In fact, the sequential nature of the path model reported in the present study brought out the impact of certain variables, such as prior knowledge, habits, and perceived health, significantly influencing other variables (perceived risk of contracting anisakiasis and past avoidance of fish consumption due to Anisakis), to impact WTP for *Anisakis*-free fish.

The importance of these findings is due to the possibility of working through interventions and practical implications on these independent variables, trying to trigger sequential processes that lead to more conscious and healthy eating behaviors. This is, therefore, a step forward in understanding how people perceive and cope with food risks, how they interpret them, and how they react to them. The results also provide information on the factors that determine the perceived risk of contracting anisakiasis and the willingness to pay for *Anisakis*-free fish. This helps to understand how previous perceived knowledge, experiences, and past behaviors guide our current behaviors and eating habits.

## 5. Conclusions

After considering the theoretical implications, some practical interventions that may arise from the results of this study can be hypothesized.

Considering the ignorance related to the knowledge of *Anisakis* and anisakiasis [59], it would be correct to undertake an awareness campaign on the issue, as well as to teach these issues through school dietary education. In fact, it has emerged from this and other studies [59] how previous knowledge is essential to avoid risky behaviors, such as, in the case of this study, the consumption of raw, unabated fish. Finally, affixing the appropriate

information (e.g., "consume after cooking" or "freeze the product before consumption") on fish packages could avoid unnecessary risks due to unawareness of the parasite, providing immediate knowledge of the same. The results showed the importance of habits, thus confirming that repeated experiences can mold food consumption behaviors: this applies to both environmentally sustainable food habits [84], and personally healthy food habits (as shown in the present contribution). Such results stress the importance of taking care of developing appropriate food experiences from childhood onward, both in the family and in school, to direct the individual future adulthood food choice towards appropriate standards (in terms of both environmental sustainability and personal health).

Similarly, promoting awareness of the phenomenon in cultures where raw fish consumption is massive (such as rural Asian populations) becomes crucial. In such populations, indeed, due to low levels of education and living situations bordering on poverty, it is not always easy to succeed in raising awareness of these issues. Poor economic resources then lead to these populations not always having easy access to prevention methods (such as culling raw fish) or optimal hygienic conditions. For this reason, to intercept the United Nations Agenda 2030 [8] goals mentioned above, massive intervention becomes necessary, especially for this large segment of the global population, which is often overlooked.

The study has some limitations: first, it is cross-sectional and therefore not totally generalizable; also, it does not allow for experimentally testing the causal nature of the links in the models. Another limitation presented by this research is the sample size which, although acceptable, could have been larger to have more statistical power. Another limitation associated with the sample is its lack of generalizability and representativeness, being composed of Italian respondents only (mostly women). This lack of generalizability may need to lead to new studies on the topic, perhaps with larger and more heterogeneous samples. Moreover, the lack of generalizability is also due to the characteristics of the sample examined, which was a convenience sample. Therefore, variability and representativeness could be increased by assuming future studies using random sampling methods, in which not only the most easily accessible people are selected.

As for possible future research, the number of participants will need to be extended, as well as considering culturally heterogeneous participants (possibly from different nations and continents). In addition, rural populations in continents such as Asia or Africa should also be reached, as mentioned above, to understand their motivational drives (dictated by their different life situations) toward risky eating behaviors. This could lead us to more carefully designed and targeted practical interventions toward such populations. Finally, the study should be repeated on the same sample over time, incorporating manipulations that might reveal actual causal links (e.g., submit to Contingent Valuation only a segment of the sample), thus incorporating both longitudinal and experimental methodological designs to bring a more solid ground to derive subsequent evidence-based interventions.

These last points are of paramount importance, as the phenomenon of anisakiasis is present all over the world [2,6,7]. Considering nationality as a control variable could highlight differences in knowledge or experience related to the parasite, trying to understand whether these differences depend on the diet or the culture and education of participants of different nationalities and cultures.

Finally, in terms of future prospects, it would also be useful to hypothesize further correlational studies that could go on to investigate how the social-psychological determinants influence our behavioral intentions. In particular, it could be possible to start with the behavior prediction models known in the literature [85,86], and eventually hypothesize a behavior prediction model specific to risky eating behaviors.

**Author Contributions:** Conceptualization, M.B., S.M., and G.A.; methodology, M.B. and G.A.; software, A.M. and O.M.; formal analysis, A.M. and O.M.; investigation, M.B. and G.A.; data curation, G.A., A.M., and U.G.C.; writing—original draft preparation, A.M., U.G.C., and L.C.; writing—review and editing, A.M., U.G.C., M.B., S.M., G.A., L.C., M.P., and O.M.; project administration, M.B., S.M., G.A., and M.P.; funding acquisition, S.M. All authors have read and agreed to the published version of the manuscript.

**Funding:** This research partly benefited from funding within the Sapienza University of Rome research project entitled "Unravelling the problem of "*Anisakis*-allergy" in humans related to seafood products: a multidisciplinary approach to improve the knowledge on the parasitic zoonosis in Italy" (protocol C26H15SBJ4, financial year 2015) and Progetto D'Ateneo 2020: "Would the investigation on the genome-wide architecture of the parasite-host interaction help to add knowledge on pathogenic features of zoonotic diseases in humans? Two cases study: the protozoan Blastocystis and the nematode *Anisakis*", under the responsibility of Simonetta Mattiucci, with the supervision of Marino Bonaiuto for the social-psychological part and the collaboration of Giulia Amicone (who was supported by a research contract for questionnaire preparation and data gathering).

**Institutional Review Board Statement:** The study was conducted in accordance with the Declaration of Helsinki and approved by the Ethics Committee of the Department of Psychology of Developmental and Socialization Processes, Sapienza University of Rome, Italy. Approval date: 24/07/17; protocol number: 759.

**Informed Consent Statement:** Informed consent was obtained from all subjects involved in the study.

**Data Availability Statement:** The data presented in this study are available on request from the corresponding author. The data are not publicly available due to ethical reasons.

**Conflicts of Interest:** The authors declare no conflict of interest.

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
