# Peer review of "Can Food Safety Practices and Knowledge of Raw Fish Promote Perception of Infection Risk and Safe Consumption Behavior Intentions Related to the Zoonotic Parasite Anisakis?"

_sustainability, doi:10.3390/su15097383_

Round 1
Reviewer 1 Report
The manuscript entitled “Knowledge, attitudes, and food safety practices of raw fish consumers: perception of risk infection and future behavior intentions can promote human health by preventing the infection with the zoonotic parasite Anisakis” deals with an interesting topic. I have mainly minor concerns, but several of them, that need to be addressed.
One of my major concerns is why didn’t you examined the relationships perceived health -> past avoidance and risk perception -> past avoidance, when they seem logical? Besides, the sampling process is questionable: it is not clear where and when you exactly collected the data online, and it is questionable that your sample is random as you state in lines 236 and 241 (in the Conclusions section you admit it). Please, specify the details of the data collection. Moreover, the first statement in the Conclusions section (lines 404-407) is not related to the examined topic, it was not even mentioned previously.
The minor issues are as follows. The title is too long, and it is rather a statement and not a title. The term “risk infection” seems to be incorrect, and in several cases, the wording is very strange (e.g., “missing topic”). The abstract should contain more specific results. The paragraph in lines 108-119 is related neither to the section title, nor to the paragraphs that follow; it would be better to place it, e.g., after the paragraph in lines 138-143. Please, revise the sentence in lines 151-153, since it is not clear. The two paragraphs in lines 187-190 and 191-193 overlap; one of them is unnecessary. In line 255, scale values (1 and 4) are missing. The citation format is not in line with the requirements when referring to more than one literature sources. There are some spelling mistakes as well, see, e.g., lines 87, 100, 289, and the top of Table 2 (df). There are unnecessary spaces in the text, see, e.g., lines 58 and 59.
Reviewer 2 Report
This is an interesting study. Here are some suggestions for your reference:
(1) The introduction is suggested to be moderately rewritten. One is to clarify the key scientific problems to be solved in this research, and then introduce several core concepts involved in the research systematically. Second, compared with the existing researches, the marginal contribution of this research is not clear, so it is suggested to further refine from the perspectives of research perspective, research content and research methods.
(2) The hypothesis is a little arbitrary. It is suggested that the author should strengthen the dialogue with classical theory. Many of the current research hypotheses are based on existing research, and do not have a dialogue with classical theory. If the literature the author refers to does not hold up, then the author's hypothesis does not hold up.
(3) How to ensure the reliability and validity of the data obtained through online survey? The extent to which the data reflect the overall picture seems unclear. Therefore, it is suggested that the author make a more detailed introduction.
(4) The discussion section needs to be further strengthened, especially the comparison of similarities and differences with similar studies.
(5) The text of Line 187-193 is repeated and needs some deletion. At the same time, many paragraphs in the text are very short, and the author is advised to merge them properly.
Reviewer 3 Report
The paper is well structured.
The author correctly used the existing literature and demonstrated the ability to use the empirical methods.
Results are clearly presented, but in discussion or conclusion could compare main results with previous studies.
There is evidence of contribution to knowledge in that area.
I suggest authors when they mentioned methods to explain the sampling methods to add1-2 lines and justify their choice.
Authors could mention shortly any ethics consideration with collection of data. As well as briefly mention the pilot study and potential changes to questionnaire,
Authors highlighted the limitations of the study and provided suggestions for further research
Additionally, the author could expand the introduction in terms of motivation. They explain the importance of topic in general but in my opinion, they can justify more their choice of this topic. While in introduction at the end of section they can mention the outline of paper and next sections.
Finally, I believe at the end of literature review is worth writing a paragraph with clear research gap and how they fill it.
Round 2
Reviewer 1 Report
Dear Authors,
The manuscript “Can food safety practices and knowledge of raw fish promote perception of infection risk and safe consumption behavior intentions related to the zoonotic parasite Anisakis?” has been developed significantly; however, some issues still apply and some of my previous comments were not (appropriately) addressed.
A new issue is that the Introduction section needs to be restructured, since with the added paragraphs the same pieces of information are repeated again and again.
The first paragraph of section 1.1 is too long and it is about at least two – although related – topics, so it is recommended to separate it into at least two paragraphs. In contrast, paragraphs in lines 212-222 are too short; they consist of only 1-1 sentence, so they should be merged logically (not necessarily in one paragraph).
The issue with the sampling method still applies; it is one thing that you used a convenience sample (this was not what I criticized previously; however, with this sample, results are not generalizable at all), but it is another thing that you state it was chosen “to increase the variability among participants and the representativeness of the population”. Convenience sample is not appropriate to increase the variability among participants nor to increase the representativeness of the population. You could increase variability and representativeness if you use random sampling; in this regard, convenience sampling is the worst choice you can make (even among not random sampling methods). It is “convenience”, so the most easily accessible persons are chosen, which guarantees neither variability, nor representativeness.
There are still strange wordings, e.g., the use of the word “anticipated” is not clear on p.3 within the given context, as well as the word “excess” in line 439.
The formatting of references is still not appropriate when more than one sources are cited. Although it is not explicitly articulated among the author guidelines, it is clear that using a hyphen between numbers means “from …. to …”, therefore, e.g., [4-2-3] is incorrect, instead, [2-4] should be used; or instead of [2-3-5], you should use [2,3,5]. Please, check any studies published in the journal.
There are still many (and even more than before) unnecessary spaces, see, e.g., lines 122, 129, 131, 141, 152, 155, 158, 163, 208, 236 (2x), and 275. On the other hand, spaces are missing from lines 165, 196, and 313. There is a further typo in line 424. Please, revise the sentence in lines 485-487. Line spacing is not appropriate in lines 623-629 and 721-724.
Reviewer 2 Report
I have no other commments, thank you.
Author Response
Thank you very much for your comments and revisions, they helped increase the quality of our manuscript.
We are glad that the revisions we made resolved your concerns.